# Integrating linguistic knowledge into DNNs: Application to online grooming detection

## Abstract

Online grooming (OG) of children is a pervasive issue in an increasingly interconnected world. We explore various complementary methods to incorporate Corpus Linguistics (CL) knowledge into accurate and interpretable Deep Learning (DL) models. They provide an implicit text normalisation that adapts embedding spaces to the groomers' usage of language, and they focus the DNN's attention onto the expressions of OG strategies. We apply these integrations to two architecture types and improve on the state-of-the-art on a new OG corpus.

## 1 Introduction

Online grooming (OG) is a communicative process of entrapment in which an adult lures a minor into taking part in sexual activities online and, at times, offline (Lorenzo-Dus et al., 2016; Chiang & Grant, 2019). Our aim is to detect instances of OG. This is achieved through binary classification of whole conversations into OG (positive class) or neutral (negative class). This classification requires the ability to capture subtleties in the language used by groomers. Corpus Linguistic (CL) analysis provides a detailed characterisation of language in large textual datasets (McEnery & Wilson, 2003; Sinclair, 1991). We argue that, when integrated into ML models, the products of CL analysis may allow a better capture of language subtleties, while simplifying and guiding the learning task. We consider two types of CL products and explore strategies for their integration into several stages of DNNs. Moreover, we show that CL knowledge may help law enforcement in interpreting the ML decision process, towards the production of evidences for potential prosecution.

Our text heavily uses slang and sms-style writing, as many real-world Natural Language Processing (NLP) tasks for chat logs. Text normalisation methods were proposed to reduce variance in word choice and/or spelling and simplify learning, e.g. (Mansfield et al., 2019) for sms-style writing. However, they do not account for the final analysis goal and may discard some informative variance, e.g. the use of certain forms of slang possibly indicative of a user category. CL analysis provides with the preferred usage of spelling variants or synonyms. We propose to use this domain knowledge to selectively normalise chat logs while preserving the informative variance for the classification task.

As demonstrated by the CL analysis in (Lorenzo-Dus et al., 2016), the theme and immediate purpose of groomer messages may vary throughout the conversation, in order to achieve the overarching goal of entrapping the victims. Groomers use a series of inter-connected "sub-goals", referred to as *OG processes* here, namely gaining the child's trust, planning activities, building a relationship, isolating them emotionally and physically from his/her support network, checking their level of compliance, introducing sexual content and trying to secure a meeting off-line. The language used within these processes is not always sexually explicit, which makes their detection more challenging. However, CL analysis additionally flags some contexts associated to the OG processes, in the form of word collocations (i.e. words that occur within a same window of 7 words) that tend to occur more frequently in, and therefore can be associated with, OG processes. We propose to exploit the relations between the OG processes and their overarching goal of OG to improve the final OG classification. We use the CL identified context windows to guide the learning of our DNN.

Our main contributions are: 1) We explore different strategies for integrating CL knowledge into DNNs. They are applied to two architecture types and demonstrated on OG detection, but may generalise to other NLP applications that involve digital language and/or complex conversational strategies. 2) The principle and several implementations of selectively normalising text through modifying a word embedding in support to classification. 3) The decomposition of conversation

analysis into identifying sub-goals. Our DNN implicitly models the relations between these sub-goals and the conversation's overarching final goal. 4) A new attention mechanism for LSTM based on the direct stimulation of its input gates, with two proposed implementations. 5) A state-of-the-art (SoTA) and interpretable OG detector. 6) A new corpus for OG detection, to be publicly released on demand, and that extends PAN2012 with more conversations and with products of CL analysis.

## 2 RELATED WORK

Villatoro-Tello et al. (2012) detected OG chat logs using a DNN to classify binary bag-of-words. This simple approach highlights the importance of commonly used words amongst groomers which we exploit for selective text normalisation. This is emphasised in (Vartapetiance & Gillam, 2014; Hidalgo & Díaz, 2012) where a set of phrases are derived from the important features of a Naïve Bayes classifier to describe common behaviours among groomers. Liu et al. (2017) obtained the current OG detection SoTA using a word embedding for semantic of important words and an LSTM.

Integrating domain knowledge into DNNs is often done with additional losses that assist with sparse and low quality data. (Muralidhar et al., 2018) penalise a DNN's output violating logical rules w.r.t. the input features. (Hu et al., 2018) use the posterior regularisation framework of (Ganchev et al., 2010) to encode domain constraints for generative models. A teacher-student architecture in (Hu et al., 2016) incorporates first-order logic rules to create an additional loss for the student network. Other works integrated prior knowledge in the design of the DNN architecture. In BrainNetCNN (Kawahara et al., 2017), the convolutions of a convolutional neural network (CNN) are defined based on the graph data's locality to account for the brain's connectivity. The training procedure may also integrate priors without modifying the DNN's architecture. Derakhshani et al. (2019) use assisted excitation of CNN neurons in the images' areas of interest, thus providing both localisation and semantic information to the DNN. An attention mechanism was used in a supervised way to focus a DNN on important words in (Nguyen & Nguyen, 2018). We experiment with these various approaches and adapt them to our domain knowledge and DNN architectures.

Linguistic knowledge was integrated to learnt word embeddings in the past. Knowledge in the form of lexicons, that carry a manual categorisation and/or ranking of words, is combined with a learnt word embedding in (Margatina et al., 2019). Three strategies are proposed, namely concatenating the lexicon and embedding features, and using the lexicon features to conditionally select or transform the word embeddings. In our study, we are concerned with a different type of linguistic knowledge. However, our modification of word embedding (Section 4.1) may also exploit this lexicon knowledge.

## 3 AUGMENTED PAN2012 DATASET

PAN2012 (Inches & Crestani, 2012) is a standard corpus for OG detection. It was gathered from Omegle (one-to-one conversations, IRC (technical discussions in groups), and the Perverted Justice (PJ) website[1] (chat logs from convicted groomers interacting with trained adult decoys), with 396 groomers and 5700 / 216,121 OG / non-OG conversations. Some non-OG chat logs contain sexual wording, making the OG classification more challenging. Conversations are truncated to 150 messages each, which limits both CL and ML analyses. To resolve this limitation, we augment the corpus with full OG conversations and the addition of new groomers from PJ, totalling 623 groomers in 6204 OG conversations (same negatives which could not be augmented to fuller conversations due to no access to the original data). Final OG / non-OG conversations total an average (std) of 215 (689) / 13 (23) messages and 1010 (3231) / 94 (489) words, respectively. Statistics on the dataset content are in the sup. materials. PJ data is freely available online and was largely used in previous social science and NLP studies, thus its use does not raise any peculiar ethical concern. For a debate on its usability see (Chiang & Grant, 2019; Schneevogt et al., 2018).

Our dataset also includes the results of a CL analysis of the new corpus using the method described in (Lorenzo-Dus et al., 2016), which involves a heavy use of manual analysis by CL experts. As part of data preparation for CL analysis, *word variants* are identified, which are either spelling variations (mistakes or intentional e.g. 'loool'→'lol'), or the same semantic meaning behind two terms (e.g. 'not comfy'→'uncomfortable'). These variants are not specific to OG, but rather reflect digital language,

---

[1] http://perverted-justice.com

and are therefore valid for other real-world chat logs. The CL analysis also identified the variants that are most used among groomers. The CL products in our dataset include: 1) the set of variants both general and groomer-preferred, 2) a set of frequent 3-word collocates (not necessarily direct neighbours, but located within a window of 7 words) that are used among many different users, and 3) a manual annotation of 2100 samples of OG processes (there are 7 types of OG processes, as identified in (Lorenzo-Dus et al., 2016) and listed in the introduction and detailed in the sup. materials) that could be associated to 3-word collocates and the context windows that these latter define. These CL products are sensitive data that might be used to help groomers refine their strategies, therefore they will only be shared on request. They are used in Sections 4-5 to train a DNN model, but this model does not require CL analysis to be performed at testing phase, as it takes raw text only as input.

## 4 METHODOLOGY

**Overarching vision and general applicability –** We integrate two CL priors into DNNs: the word variants and the identification of OG processes. Word variants provide knowledge of same semantic meaning, which allows reducing variance in the text. The knowledge of groomer's preferred variants brings an implicit and selective text normalisation that supports the classification task. It is achieved through a reduction of distances between non-discriminative variants in a word embedding. This selective normalisation is applicable to other classification tasks from real-world chat logs, provided an updated selection of the preferred and discriminative variants. As highlighted in Section 3, the variants reflect digital language and are relevant to different analyses of chat conversations. The selection of discriminative variants is done easily and automatically following a procedure described in Section 4.1 using empirical occurrences in positive and negative conversations. This knowledge integration is also applicable to all DNNs that use a word embedding to capture word semantic.

The use of OG processes aides in differentiating between causal conversations involving sexual language, and OG conversations with complex strategies and sub-goals (i.e. OG processes). The language associated to OG processes, reflected by the 3-word collocates and context windows that they define, may be more informative than traditionally used simple sexual wording in making this distinction. We propose 3 strategies to integrate this knowledge, namely the definition of sub-tasks and two stimulations of DNN attention. They all guide the learning by providing focus on contexts of interest (a valuable complement to attention mechanisms, as demonstrated in our experiments), and by implicitly modelling the relation between sub- and final goals. This CL knowledge integration principle is generally applicable to the analysis of complex conversations, provided an appropriate CL identification of the conversation's sub-goals and of their associated language through context windows. This identification of sub-goals has been the focus of many social science studies. For example, a large corpus of works have identified strategies for persuasion and manipulation in extreme ideology groups, e.g. (Brindle, 2016; Nouri & Lorenzo-Dus, 2019; Lorenzo-Dus & Nouri, 2020; Saridakis & Mouka, 2020) for radical right hate speech and (Baker et al., 2021) for jihadi radicalisation. This established baseline of knowledge may be integrated into DNNs in multidisciplinary works. The identification of frequent 3-word collocates is automated, as described in (Lorenzo-Dus et al., 2016). The association of their occurrences to the identified sub-goals is the only task that may require additional manual work. Our stimulation of DNN attention may also be more generally used to focus a DNN's attention on *a priori* known important elements of a training set.

**Base models –** We demonstrate the general applicability of our CL integration strategies by applying them to two DNN architecture types representative of the two NLP standards of recurrent and transformer models. The recurrent DNN of Liu et al. (2017) is the current SoTA for OG classification. It comprises a language model that builds two word and sentence embeddings, and an OG classifier with two LSTMs and a linear projection. Our *base model #1* is a modified version (Fig. 1 left) with the word embedding provided as input to the OG classifier in place of the sentence embedding. This word embedding will be more directly impacted by our CL integration, and it increases explainability as will be seen next. It may be replaced by similar embeddings, and we also present results using the pre-trained GloVe (Pennington et al., 2014). Further, to compensate for the loss of sentence structure modelling previously provided by the sentence embedding, and to account for the longer sequences of inputs into the classifier, we add an unsupervised attention mechanism (Luong et al., 2015) into the classifier. Following the method in (Luong et al., 2015), the hidden states of the last LSTM for all words of the conversation are provided to the attention mechanism that outputs a conversation embedding of the same size as the LSTMs hidden state, namely 256.

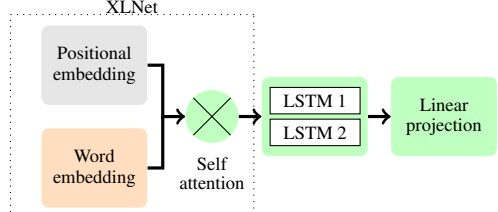

Figure 1: Base models #1 (left) & #2 (right), for integration of CL priors. Orange and green indicate where word variants and OG processes priors may be integrated, respectively.

XLNet (Yang et al., 2019) is a popular transformer model, a SoTA for many NLP tasks, and therefore a strong baseline for this study. It iteratively refines word embeddings, starting from an initial embedding that captures word semantic similarly to that of Liu et al. (2017), and attaining richer word representations that account for word relationships within a sentence using a positional embedding and self attention layers. The refined contextualised word embeddings are classified by linear projection. In our application, this projection fails to handle our class imbalance and always outputs the same class with F-score at 0.392. Providing the contextualised word embeddings to a two-layer LSTM, whose last hidden state is used as a conversation embedding to be classified by the linear projection, solves this issue[2] and forms *base model #2* (Fig. 1 right). The combination of a transformer model with LSTM is not new, see for example (Ma, 2019), and has the advantage of allowing the use of our LSTM-based knowledge integration strategies (see 'Stimulating LSTM input gates' in Section 4.2).

**Input to the models –** The analysis is performed on whole conversations, and the final OG / non-OG classification is obtained for the whole conversation, rather than per-message. Messages are separated by the [SEP] token, so that inter-text representations can be modelled. Messages from both users are included with no distinction. For base model #2, the [CLS] token is added at the beginning of conversations following the XLNet standard. Conversations longer than 2,000 words are truncated to retain their end part (12% / 8e-5% of OG / non-OG conversations). All base and CL-augmented DNNs take raw text as input only. The only text preparation prior to the DNN is tokenisation of named entities. We do not apply explicit text normalisation such as (Mansfield et al., 2019) as part of text preparation, since the methodological premise of the paper is the design of a hybrid approach where an ML model incorporates its own text normalisation informed by CL knowledge.

### 4.1 IMPLICIT AND SELECTIVE TEXT NORMALISATION BASED ON WORD VARIANTS

The natural stage of DNNs to integrate knowledge on word variants is the word embedding that captures word semantic (i.e. before any LSTM or self-attention layer in the base models). The mean occurrence frequency of variants in the OG corpus is significantly larger, by two orders of magnitude, than that of all words (see sup. materials). Therefore, using these common words to modify the word embedding may have a strong impact on classification. We propose 3 strategies to modify the embedding based on our set of $N$ pairs of variants $\{(v_1^i, v_1^j), ..., (v_N^i, v_N^j)\}$ using the principle that *words with same semantic should be moved closer to each other in the embedding space*.

Although variants have same intended meaning, some may be discriminative of groomers' language. Hence, it may be useful for OG classification to keep them apart in the word embedding space. The significance of word $w$ for classification is determined based on empirical occurrences in OG and non-OG conversations within the training set: $\delta p(w) = |p(w|y_{pos}) - p(w|y_{neg})|$. If $\delta p(v_k^i)$ or $\delta p(v_k^j)$ are high (i.e. within the last 5 percentiles for all words), *we do not use the pair for modification of the embedding*. We considered increasing the separation between these variants, but we found that this modifies too much the embedding and reduces its semantic representation power. Out of 4590 pairs of variants, we retain 2955 for modification of the word embedding. In effect, this selective modification applies an implicit and selective text normalisation which supports the OG classification.

We experiment with 3 implementations that may apply to different usage scenarios such as training a new language space (supervised word embedding modification), or modifying an existing one before training a new classifier (manifold-based) or before fine-tuning an existing classifier (elastic pulling).

---

[2]The reason for this behaviour remains to be investigated.

**Supervised word embedding modification –** A regularisation term is added with weight $\lambda$ to the language modelling loss $\mathcal{L}_{Emb}$ to minimise the $L_2$ distance $\mathcal{D}$ between the selected word variants' embeddings: $\tilde{\mathcal{L}}_{Emb} = (1 - \lambda) \mathcal{L}_{Emb} + \lambda \left[ \frac{1}{N} \sum_{k=1}^{N} \mathcal{D}(v_k^i, v_k^j) \right]$ .

**Manifold learning –** We perform a global transformation of the existing word embedding by building a new space through manifold learning from an edited pairwise distance matrix with $\tilde{\mathcal{D}}(v_k^i, v_k^j) = \lambda \mathcal{D}(v_k^i, v_k^j)$ where $\lambda \in [0, 1]$ modulates the strength of distance reduction between selected variants. We use Robust Diffusion Map (Paiement et al., 2014), but other manifold learning methods could be explored. This implementation requires re-training subsequent modules from scratch, as words' new embeddings may be very different from initial ones. Note that we make an unusual use of manifold learning for word embedding modification rather than for dimensionality reduction. It is possible to reduce dimensionality, which may help to combat overfitting for classification, as discussed in (Yin & Shen, 2018). However, the dimensionality of the word embedding is unchanged in our experiments.

**Elastic pulling –** Our third implementation modifies the existing word embedding 'in place' through local movements that pull together the representations of selected variants. This mostly preserves all words' original representations (i.e. coordinates in the embedding space), thus limiting the amount of change needed for the classifier to simple fine-tuning. Two variants' representations $v_k^i$ and $v_k^j$ of coordinates $\mathbf{x}_k^i$ and $\mathbf{x}_k^j$ are pulled towards their centre $\hat{\mathbf{x}}_k = \frac{\mathbf{x}_k^i + \mathbf{x}_k^j}{2}$ by the amount $\delta_{\mathbf{x}_k^i} = \hat{\mathbf{x}}_k - \mathbf{x}_k^i$ modulated by $\lambda \in [0, 1]$: $\tilde{\mathbf{x}}_k^i = \mathbf{x}_k^i + \lambda \delta_{\mathbf{x}_k^i}$. We propagate the pull operation to neighbouring word representations, with strength of pull decreasing with distance (i.e. modulated by a radial basis function (RBF) $\phi_k^i$ centred on $\mathbf{x}_k^i$), so as to preserve the pairwise relationships between variants and their neighbours: $\tilde{\mathbf{x}} = \mathbf{x} + \lambda \phi_k^i(\mathbf{x}) \delta_{\mathbf{x}_k^i}$. We use an inverse multiquadric $\phi_k^i(\mathbf{x}) = (||\mathbf{x} - \mathbf{x}_k^i||^2 + \gamma^2)^{-\frac{\beta}{2}}$, with global support so that all words can be considered for propagating each pulling operation without the need for a costly identification of those word representations that are located within the pulling's neighbourhood. $\beta$ and $\gamma$ tune the RBF's decay rate i.e. the locality of propagated pull. We found that the method is not very sensitive to these values as long as the pull's reach is sufficient, within a radius of the order of magnitude of $\delta_{\mathbf{x}_k^i}$, and we set them empirically to 1.0 and 3.0 respectively.

## 4.2 INTEGRATING KNOWLEDGE ON OG PROCESSES

Our annotated samples of OG processes are associated to 3-word collocates, which are used to identify contexts of interest. We define a continuous representation of the presence of the 7 OG processes using 7 Gaussian Mixture Models (GMM) with components centred on the 3-word collocates and their std being the span of each collocate (max. 7 words as mentioned in Section 3). We propose 3 uses to focus the attention of the DNN on parts of the conversations that implement the OG processes and on the associated language, and to implicitly model the relations between OG processes and OG.

**Auxiliary OG process detection –** A second output branch is added to the DNN after the LSTMs, with a fully-connected layer and MSE loss, to estimate the pseudo occurrence probability of the 7 OG processes provided by their GMMs, at each word location. For base model #1, we experimented with adding an attention block as in the main branch, but found that this didn't help with the OG process detection, probably because this task is more local and doesn't need to consider as large a context as classification of whole conversation. The new branch also serves as additional regularisation to prevent overfitting given the class imbalance between (non-)OG chat logs. Further, it allows for an OG process-based interpretation of what the DNN considers as relevant clues for OG classification.

**Stimulating attention –** Both the unsupervised attention of Luong et al. (2015) in base model #1, and the self-attention layers of base model #2, compute an attention energy $e_t$ for word at position $t$. It may be stimulated during training to guide the DNN's attention on occurrences of OG processes[3]. We propose two strategies that are not mutually exclusive and may be combined: a) through **supervision** by the sum $\mathcal{G}$ of GMMs used as ground-truth distribution of the salient locations and attention energies: $\mathcal{L}_{\text{attention}} = \frac{1}{T} \sum_{t=1}^{T} (e_t - \mathcal{G}(t))^2$ , with $T$ the length of messages from both users. This is similar to (Nguyen & Nguyen, 2018), but with $\mathcal{G}$ highlighting higher-level OG processes rather than single important words. b) through direct **excitation** of the attention energies, inspired by

---

[3] No annotation of OG processes (i.e. GMM) is required at testing time.

(Derakhshani et al., 2019) that excited CNN's activations to speed up localisation learning in images. We propose two possible implementations: **(A)** $\tilde{e}_t = e_t + \mathcal{G}(t)\, e_t$, and **(B)** $\tilde{e}_t = e_t + \mathcal{G}(t)$.

**Stimulating LSTM input gates –** An alternative (or complement) to stimulating an attention mechanism is to stimulate LSTM cells directly, during training[3], in locations containing OG processes indicated by $\mathcal{G}$. This is a new way to stimulate attention and to encourage the LSTM to recognise and focus on the contexts of OG processes. We propose two implementations that are not mutually exclusive and may be combined: a) through **supervision** by minimising the loss between the average input gates' activations $i_t$ and the combined GMMs: $\mathcal{L}_{\text{stimulation}} = \frac{1}{T}\sum_{t=1}^{T}(i_t - \mathcal{G}(t))^2$ ; b) through **excitation** of activations, following the same idea as for exciting attention. The input gate activation $i_t^d$ of each LSTM cell $d$ is augmented during OG processes, indicated by a peak of $\mathcal{G}$, through: $\tilde{i}_t^d = i_t^d + \frac{\mathcal{G}(t)}{D}\sum_{d=1}^{D} h_t^d$ . We average over the hidden states $h_t^d$ of the $D$ LSTM cells in the layer, by analogy to (Derakhshani et al., 2019) that averaged over all channels of a CNN's activation map.

## 5 EXPERIMENTS

Both original and modified base model #1 (including its word embedding) are trained from random weights on our dataset. Experiments with the GloVe embedding use GloVe's pre-trained weights from Common Crawl 840B (Pennington et al., 2014). The XLNet part of base model #2 is pre-trained on BookCorpus and English Wikipedia, see (Yang et al., 2019). Detailed training procedures are provided in the sup. materials. For base model #2, the selective text normalisation is not tested due to lack of time in fine-tuning XLNet. The attention is stimulated in the final self-attention layer only, and future experiments may test other locations.

We divide the corpus into 30% of users for training, 70% for testing, and 30% of training for validation, using a similar ratio as Inches & Crestani (2012). This division based on users ensures that the model may not recognise the specific language of a groomer, but focuses on trends in OG language. OG classification is evaluated by: precision, recall, area under precision-recall curve (AUPR), $F_1$ score, and the $F_{0.5}$ score used in (Inches & Crestani, 2012) to weight the precision metric higher. The effects of selective text normalisation are further measured by their proportion of distance reduction between selected variants $\Delta\mathcal{D} = \frac{1}{N}\sum_k \frac{|\mathcal{D}(v_k^i, v_k^j)^{new} - \mathcal{D}(v_k^i, v_k^j)^{old}|}{\mathcal{D}(v_k^i, v_k^j)^{old}}$ and average resulting distance $\overline{\mathcal{D}} = \frac{1}{N}\sum_k \mathcal{D}(v_k^i, v_k^j)$. Accuracy is provided in the sup. materials.

### 5.1 EVALUATIONS OF THE INDIVIDUAL CL-KNOWLEDGE INTEGRATION STRATEGIES

We evaluate the individual effects of the different CL augmentations in Table 1 in comparison to non-augmented models. We also try combining the two supervised and excitation-based methods for stimulating attention and LSTM input gates. For a fairer comparison of the selective text normalisations, their modulation parameters $\lambda$ are approximately adjusted to provide a loosely similar $\Delta\mathcal{D}$, as reported in the sup. materials.

Base model #1 (before augmentations) obtains similarly good results using both word embeddings, even though GloVe encompasses more words in a larger embedding, with resulting larger $\overline{\mathcal{D}}$. Base model #1 generally responds well to all CL-integrations (with some implementations of the selective text normalisation performing better than others, as discussed next). The selective text normalisation is less effective on GloVe, maybe due to a more drastic reduction of its larger initial distances. A grid search on $\lambda$ may be performed in the future to investigate this behaviour. Base model #2 outperforms base model #1, which confirms XLNet's status as one of the NLP SoTA. It also responds well to augmentations based on knowledge of OG processes, with all metrics consistently improved. Thus, the selective text normalisation for XLNet's embedding for word semantic (i.e. before self-attention layers) remains an interesting strategy to evaluate in future experiments.

Among the 3 proposed implementations of selective text normalisation, only the pulling version provided an improvement, while the other two hindered OG classification in spite of a smaller reduction of distances between selected variants. For the supervised approach, this may be explained by the new loss term conflicting with the original word embedding loss. For manifold learning, although the algorithm preserves pairwise distances by design (as verified in sup. materials), this does not seem enough to fully preserve the semantic representation power of the word embedding.

Table 1: Impact of each CL augmentation on OG classification. Bold are improved results with respect to no augmentation, i.e. base models.

| Model | Strategy | | Precision | Recall | AUPR | $F_1$ | $F_{0.5}$ | $\Delta\mathcal{D}/\overline{\mathcal{D}}$ |
|---|---|---|---|---|---|---|---|---|
| #1 | No augmentation | | 0.867 | 0.794 | 0.867 | 0.829 | 0.851 | – / 3.72 |
| | Supervised word embed. modif. | | 0.834 | 0.765 | 0.824 | 0.798 | 0.819 | 0.75/0.91 |
| | Manifold learning | | 0.849 | 0.723 | 0.809 | 0.820 | 0.781 | 0.65/1.29 |
| | Elastic pulling | | 0.878 | 0.808 | **0.877** | **0.841** | **0.863** | 0.83/0.61 |
| | Aux. OG process detection | | 0.890 | 0.768 | **0.873** | 0.825 | **0.863** | – |
| | Stim. attention | supervised | 0.839 | 0.804 | **0.879** | 0.821 | 0.832 | – |
| | | excitation (A) | 0.822 | 0.817 | **0.877** | 0.820 | 0.821 | – |
| | | excitation (B) | 0.870 | 0.808 | **0.906** | **0.838** | **0.857** | – |
| | | superv.+excit. (A) | 0.859 | 0.819 | **0.897** | **0.838** | 0.851 | – |
| | | superv.+excit. (B) | 0.929 | 0.741 | **0.908** | 0.824 | **0.884** | – |
| | Stim. LSTM | supervised | 0.924 | 0.752 | **0.891** | 0.829 | **0.883** | – |
| | | excitation | 0.856 | 0.797 | 0.863 | 0.825 | 0.843 | – |
| | | superv.+excit. | 0.906 | 0.781 | **0.901** | **0.839** | **0.878** | – |
| #1 w. GloVe | No augmentation | | 0.879 | 0.789 | 0.861 | 0.832 | 0.860 | – / 8.90 |
| | Supervised word embed. modif. | | 0.868 | 0.739 | 0.834 | 0.798 | 0.839 | 0.90/0.93 |
| | Manifold learning | | 0.896 | 0.708 | 0.839 | 0.791 | 0.851 | 0.85/1.23 |
| | Elastic pulling | | 0.880 | 0.772 | **0.862** | 0.823 | 0.856 | 0.92/0.73 |
| #2 | No augmentation | | 0.900 | 0.871 | 0.940 | 0.886 | 0.894 | – |
| | Aux. OG process detection | | 0.918 | 0.861 | **0.943** | **0.889** | **0.906** | – |
| | Stim. attention | supervised | 0.919 | 0.862 | **0.994** | **0.890** | **0.907** | – |
| | | excitation (A) | 0.894 | 0.885 | **0.945** | **0.889** | 0.892 | – |
| | | excitation (B) | 0.916 | 0.866 | 0.940 | **0.891** | **0.906** | – |
| | | superv.+excit. (A) | 0.891 | 0.881 | **0.941** | 0.886 | 0.889 | – |
| | | superv.+excit. (B) | 0.918 | 0.862 | **0.941** | **0.889** | **0.906** | – |
| | Stim. LSTM | supervised | 0.938 | 0.857 | **0.944** | **0.896** | **0.921** | – |
| | | excitation | 0.896 | 0.896 | **0.944** | **0.887** | 0.892 | – |
| | | superv.+excit. | 0.960 | 0.846 | **0.945** | **0.899** | **0.935** | – |

On the other hand, the more gentle elastic pulling could preserve the original semantic representation of the word embedding while introducing an implicit normalisation of the selected word variants that supports OG classification.

It is worth noting, for base model #1, that $\overline{\mathcal{D}}$ the average distance between the representations of two selected variants is at 3.72, higher than the average distance between all other pairs of words which is at 2.86. Therefore, base model #1, even though fully trained on OG classification, was not able to discover on its own the knowledge that some variants have same semantic meaning while not being discriminative for the OG classification task, and could therefore have same or similar representations. This, together with the improved results from modifying the word embedding, demonstrate the usefulness of integrating this knowledge into the model.

All integrations of knowledge on OG processes improved the performance of the models. This demonstrates that focusing the DNNs' attention on the language associated with OG processes does help capturing subtleties of grooming language. In addition, when exploring the attention energies of (non-augmented) base model #1, we observe that the contexts that the model learnt to focus on are not related to our labelled instances of OG processes: the average (std) attention energy for these instances is 0.0009 (0.0002), lower than energy across all conversations at 0.0016 (0.0128). A similar observation is made for base model #2, where tokens' energies are obtained from the last self-attention layer similarly as in (Sood et al., 2020) by retaining the max pairwise energy for each token (row) and normalising by the sum of retained energies. This is done for each attention head, before averaging across heads. The resulting average (std) energy for our instances of OG processes is 0.110 (0.072), slightly lower than the energy across all conversations at 0.120 (0.088). Thus, neither models were able to discover on their own the sub-goals that the CL analysis of Lorenzo-Dus et al. (2016) identified, and their associated language. This knowledge is therefore an added value for the models, as also demonstrated by the improved results. The 3 strategies seem roughly equally helpful at focusing the DNN's attention and capturing the subtleties of grooming language, and future work

Table 2: Comparative evaluation of OG classification methods

| Method | Precision | Recall | AUPR | $F_1$ | $F_{0.5}$ |
|---|---|---|---|---|---|
| Naive Bayes | 0.240 | **0.974** | 0.727 | 0.385 | 0.283 |
| SVM | **0.997** | 0.337 | 0.748 | 0.504 | 0.716 |
| Decision Tree | 0.693 | 0.642 | 0.637 | 0.667 | 0.682 |
| Random Forest | 0.987 | 0.400 | 0.718 | 0.569 | 0.763 |
| Liu et al. (2017) | 0.919 | 0.735 | 0.885 | 0.817 | 0.875 |
| BERT | 0.837 | 0.711 | 0.815 | 0.711 | 0.808 |
| Base model #1 | 0.867 | 0.794 | 0.867 | 0.829 | 0.851 |
| Base model #1 + L1 Regularisation | 0.880 | 0.759 | 0.857 | 0.815 | 0.853 |
| Base model #1 + L2 Regularisation | 0.896 | 0.783 | 0.890 | 0.835 | 0.871 |
| Base model #2 | 0.900 | 0.871 | 0.940 | 0.886 | 0.894 |
| Base model #2 + L1 Regularisation | 0.885 | 0.881 | 0.940 | 0.883 | 0.883 |
| Base model #2 + L2 Regularisation | 0.913 | 0.865 | 0.941 | 0.888 | 0.903 |
| Augmented model #1 | 0.930 | 0.777 | 0.924 | 0.847 | 0.895 |
| Augmented model #2 | 0.953 | 0.853 | **0.948** | **0.900** | **0.931** |

will explore their combinations. Improvements are more consistent for AUPR and precision (and consequently $F_{0.5}$), thanks to fewer false positives. This reduction in false positives may be due to an easier distinction of OG conversations from neutral but sexually-oriented ones.

For both stimulation strategies (attention mechanism and LSTM input gates) combining the supervision and excitation approaches provides better results than using them individually. This suggests that these two processes support each other during optimisation. Indeed, improved DNN's attention (expressed in $e_t$ and $i_t$) from excitation may assist with the supervised attention task. In addition, improved attention from supervision may also reinforce the excitation and allow it to work at its best.

## 5.2 OG CLASSIFICATION PERFORMANCE

The prior integration methods are combined into fully augmented models #1 and #2. All algorithms for selective text normalisation have the same aim, thus we only retain the best performing elastic pulling. As suggested by the previous discussion on the supervision-excitation symbiosis, the different strategies and their implementations for integrating knowledge on OG processes may be complementary. Thus, we use all 3 strategies, combining supervision and excitation for both stimulation strategies, and choosing excitation B over A due to its better results. For augmented model #2, only the augmentations tested individually in Table 1 are used. Comparative results on OG classification are provided in Table 2 over baselines and SoTA NLP models.

Although base model #1 does slightly worse than (Liu et al., 2017), its augmented version outperforms it by a margin. XLNet of base model #2 is the best performing of non-CL-augmented models. Its augmentation by the combined integration of CL knowledge on word variants and OG processes significantly improves its performance and produces the new SoTA.

For both base models, the combined augmentations (Table 2) add up to improvements that are superior to those of individual augmentations (Table 1). This is particularly true for augmented model #1 that accumulates all 4 augmentations, while augmented model #2 is limited to 3. Its SoTA may be further improved in the future through adding the selective text normalisation. In the sup. materials, we further explore how the individual augmentations add up through their progressive additions to a simple LSTM model. We observe that their respective benefits are complementary.

In order to verify that the improved results do come from a better understanding of language provided by CL knowledge, rather than merely from additional regularisation, we also compare against L1 and L2 regularised version of both base models. Although regularisation does improve the results, the performance gains from integrating CL knowledge are significantly superior for both models.

**Visualisation –** Since the augmented models make use of OG processes' recognition to capture the language associated with grooming, their auxiliary OG process detection may be used to highlight, at word level, those parts of the conversation that the model associates to OG processes. These are visualised in Fig. 2, where the estimated likelihood of the *Compliance testing* process is indicated in shades of red. The DNN focused on questions about personal situation and on invitations to talk

crazy4justin06: u been in school long ?
tennisboy213: kinda
tennisboy213: so is ur mom home ?
crazy4justin06: not yet
tennisboy213: do u wan na talk on the phone
crazy4justin06: r u gon na answer this time ?
tennisboy213: yup
crazy4justin06: how did u know it was me tat called ?

tennisboy213: the number seemed like it was out of town
tennisboy213: so r u gon na call now ?
tennisboy213: hey i can call u
tennisboy213: that way u won't get charged for long distance
crazy4justin06: i got a calling card
tennisboy213: its ok if i call u we can talk longer

Figure 2: Visualisation of detecting the *Compliance testing* OG process.

over the phone, as indicators of compliance testing happening, in line with our general understanding of this OG process. While these elements of discussion may seem neutral enough and may not be captured by generic OG classifiers, the DNN's understanding of OG processes and of their relation to OG made it increase the OG classification score at each detection of OG process. In future work, a similar visualisation could be performed using the attention energies $e_t$ and LSTM input gate activations $i_t$, to assess the better capturing of subtle language clues provided by the two other strategies for integrating knowledge on OG processes, and their effect on OG classification.

## 6 CONCLUSION

We have explored the integration of CL knowledge in a hybrid (data- and knowledge-driven) DNN. We considered two types of CL knowledge, namely: 1) variants of semantically equivalent words that are, or not, discriminative of OG, and which we use to perform a selective text normalisation in support of classification. Existing normalisation methods would apply to full text with no such distinction, thus failing to provide this support. 2) The identification of some OG processes and their associated language, which we use to focus the attention of the DNN on subtle language clues. We compared several integration approaches, including a new method for stimulating an LSTM's attention directly through its input gates, without the need for an external attention mechanism. For our final augmented model, we selected a gentle pulling method for selective text normalisation as well as a combination of auxiliary tasks, and supervision and excitation for stimulated attention and stimulated LSTM, whose benefits add up to produce the SoTA. We demonstrated the general applicability of our approaches on two architecture types of recurrent and transformer neural networks and two word embeddings of different complexities. Our results show performance improvements over base and SoTA models for both architectures, while allowing for a CL based interpretation of the classification decision through visualisation of predicted OG processes and DNN's attention.

While we have demonstrated the applicability of these methodologies for CL knowledge on OG, we see the potential for other domains that utilise similar representations or model architectures (see discussion in Section 4). The selective text normalisation that we propose is more generally applicable to other classification tasks from chat conversations, and its proposed implementations are usable on other DNNs that use a word embedding to capture word semantic. The decomposition of conversations into sub-goals may be obtained from CL studies on other applications. Some of the proposed strategies may also allow the integration of other (non-CL) domain knowledge. It may be generally useful to estimate auxiliary quantities that are known to be relevant to the task and that may usefully constrain the DNN's attention and learnt features. Our two other methods for focusing the DNN's attention (i.e. stimulated attention and stimulated LSTM input gates) may be generally used to weight more some important elements of the training data.

**Perspectives for OG prevention –** The proposed OG classification method has been designed based on requirements from specialised law enforcement to assist in the investigation of large quantities of chat logs. Its intended usage is to facilitate triage by law enforcement of digital materials that could be seized from suspected offenders after enough evidence allowed launching the procedure. Flagged conversations are to be investigated more thoroughly by a trained human operator following law enforcement's strict robustness and security protocols to ensure integrity. Within this usage scenario, there is, therefore, no risk of innocents to be automatically prosecuted. The aim of our work is not to address the possible biases in the human decision, which are addressed by law enforcement's protocols. However, the proposed visualisation that helps focusing on key aspects of the conversation, together with the reduced workload and associated lowered time pressure, may allow a more thorough and fairer investigation of the flagged conversations. Mitigation measures should be put in place, but these are outside the scope of this work.

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
