# OpenReview forum: "Integrating linguistic knowledge into DNNs: Application to online grooming detection"
_ICLR.cc/2021/Conference — Reject_

### Official Review · AnonReviewer3 · 2020-10-16
**Some good ideas, but needs more work**

**Rating:** 4
**Confidence:** 4

**Review:**

##########################################################################

Summary:

This paper proposes two families of methods for integrating external knowledge
into neural networks aimed at classifying instances of online grooming. The
first family focuses on incorporating knowledge of word semantic similarity into
their representations. The second, on different attention mechanisms for
incorporating knowledge about theoretical stages of online grooming.

The authors perform a solid amount of experiments to assess the differences
between their suggested methods and baseline models.


##########################################################################

Reasons for score:

However, the way the paper is written and structured make it difficult to
understand. While looking at the results, it is hard to really assess the
contribution of each suggested strategy, and how they compare to each other.

Finally, the paper presents some serious conceptual gaps that undermine its
overall credibility.

##########################################################################

Pros:

- Fair amount of experiments for assessing impact of each proposed strategy.
- The ideas for modeling different word variants could be built upon. I
  especially like the idea of Elastic Pulling for combining the semantic
  information of word representations. This idea would be useful to the research
  community focusing on combining word representations.


##########################################################################

Cons:

- Some serious conceptual gaps, and wrong claims. .
- The paper is overall unclear and difficult to understand.
- The task the paper is addressing is not well specified.
- The methods are also not clearly explained.
- Results are difficult to interpret and analyses are lacking.
- Lack of ethical considerations for a system that could be used in law
  enforcement. I would have liked to see a more detailed discussion on the
  societal implications that systems aiding law enforcement could have, and in
  particular, which measures should be taken for avoiding the prosecution of
  innocent people by systems like this.

#########################################################################

Comments and suggestions for the authors:

- The term "text normalisation" is spread throughout the paper, but is never
  concretely defined, and is not immediately inferable.

- I was not familiar with the "word semantics representation" noun phrase. After
  reading the paper I am pretty sure that you mean "vector space model", or word
  embeddings. Why not use these terms that will probably be more familiar to
  potential readers?

- In Figure 1 (left), you wrote Embedding; at the right you wrote WSR. Are these
  equivalent?

- I understand that you are doing classification at the conversation level, but
  in page 3, in the "Base models" paragraph, you mention that "with the WSR's
  embedding provided as input to the OG classifier in place of a sentence
  embedding". In order to classify a conversation, you need a vector
  representation of it. How is this obtained? In other words, how are you
  aggregating the contextualized word representations (i.e., the output of the
  LSTM or the XLNet encoder), into a single vector representation of the
  conversation? Are you using the last hidden state of the LSTM or a pooling
  method? Are you using the [CLS] token of XLNet or something else?

- You mentioned that your dataset contains full conversations with an average of
  431 messages per conversation. Are all the conversation turns separated by the
  [SEP] token? What is the average message length? What max input length did you
  use as a hyperparameter? Did you use the same text input for Model 1 and Model
  2?

- Saying XLNet is the SoTA for NLP is a false statement (p. 3 second-last
  paragraph). First of all, NLP encompasses several tasks and there is no single
  model superior to all the others in every task. Second, XLNet has already been
  beaten in several tasks. See the Glue benchmark
  (https://gluebenchmark.com/leaderboard) for a few examples. You mention that
  XLNet is the SoTA of NLP again in p. 6; sec. 5.1; second paragraph.

- Saying that XLNet iteratively refines word embeddings from a WSR similar to
  that of LIU et al. (2017) is only tangentially true and is misleading, in my
  opinion. Recurrent models such as those relying on LSTMs, are profoundly
  different to those based on transformers such as XLNet.

- I am not sure that there is a clear correspondence between using recurrent
  models such as LSTMs and better handling of class imbalance. How is a
  two-layer LSTM going to help handling class imbalance?

- You mentioned several times that "text normalisation" was a strong point of
  your contribution, but in the last paragraph of section 4 (p.4) you say that
  you do not apply "text normalisation". I understand that you might be
  referring to different kinds of normalisation, but I think this terminology is
  confusing. I suggest using more precise language to clearly differentiate
  what you are referring to.

- What do you mean by "intersection tests"?

- I suggest using the word "representations" rather than "coordinates" for
  referring to the vector representation of a word (p. 4; elastic pulling
  paragraph).

- The correspondence between rows in Table 1 and the strategies discussed in
  section 4.2, are not immediately apparent.

- When not using GloVe embeddings, how did you initialize your word
  representations?

- You mention in Table 1 that bold are improved results, but improved with
  respect to what?

- You report precision and recall in Table 2, but not in Table 1. I think it
  would be valuable to include these in Table 1, rather than the distance
  reduction and average resulting distance metrics which are only valid for a few
  strategies.

- You could check the following papers for more context on combining
  representations of different word variants:
  * [Attention-based Conditioning Methods for External Knowledge Integration](https://www.aclweb.org/anthology/P19-1385/)
  * [Compositional-ly Derived Representations of Morphologically Complex Words in Distributional Semantics](https://www.aclweb.org/anthology/P13-1149/)
  * [Composition in Distributional Models of Semantics](https://onlinelibrary.wiley.com/doi/full/10.1111/j.1551-6709.2010.01106.x)


#########################################################################

Some typos:

- p. 4; Manifold Learning - "by building, using manifold learning, a new space"
  could be better written as "by building through manifold learning a new space"
  or "by building a new space through manifold learning"

- p. 4 - "it is possible reduce dimensionality" -> "it is possible to reduce
  dimensionality"

- p.5, sec. 4.2, first paragraph - "collocates and std the span". Not sure what
  std is supposed to mean here.

---

> ### Author Response · Authors · 2020-11-16
> **Response to reviewer - part 1**
>
> We thank the reviewer for their insights. We are sorry that the reviewer finds the methods and discussions of experimental results hard to understand. In the revised paper, we have expanded (as much as page limit allows) the descriptions and discussions to alleviate this issue. We especially expanded the discussions around the ablation study, which evaluates and compares each suggested strategy. Next, we respond in turn to their raised questions and comments. We hope that this will address the criticism about "conceptual gaps". If not, we would appreciate a more explicit description of these gaps, to allow us to address them.
>
> - *The task the paper is addressing is not well specified.*
>
> The task is the classification of conversations between instances of online grooming (positive class), and neutral (negative class) conversations. In the revised submission, we have expanded the second sentence of the paper (originally: “The aim of this work is to detect instances of OG through classification of whole conversations.”) to clarify this.
>
> - *Lack of ethical considerations*
>
> As discussed in the introduction and in the dedicated discussion in Section 6 “Perspectives for  OG  prevention”, the system was co-developed with specialised law enforcement, and its “intended usage is to facilitate triage by law enforcement”. It is not intended to replace a human decision. Within this usage scenario, there is, therefore, no risk of innocents to be automatically prosecuted, since this decision is always made by a trained police officer after reviewing the conversation.
>
> In fact, not all courts would accept linguistic evidence. Our law enforcement partners only aim at using this system when they already have enough evidence, as part of their toolkit to analyse large quantities of seized digital materials including conversations.
>
> The co-development of the system with law enforcement ensures that their own security protocols will be followed throughout the usage of the system, to preserve integrity. In particular, the system would not be provided to any police officer, but only to trained operators, using robust mechanisms that are already being in use by law enforcement in the context of OG investigations. Therefore, while a human review of the flagged conversations might in general suffer from biases and prejudices, security protocols are already in place to mitigate these natural human biases.
>
> The aim of our work is *not* to address the possible biases in the human decision. However, our method may indirectly help in achieving a fairer handling of the flagged conversations. Indeed, the visualisation provided by our method eases the analysis of key parts of the conversation. Additionally, the triaging would allow operators to focus on a few flagged conversations while spending less time on the others. This reduced workload and associated lowered time pressure, associated with the visualisation, may allow a more thorough and fairer investigation.
>
> The “Perspectives  for  OG  prevention” section has been expanded in the revised manuscript to reflect this discussion.
>
> While one may argue that our system may be diverted from its intended use and deployed more largely online without following the law enforcement’s robustness and security protocols, we wish to point out that this is the case for many AI systems, and therefore that this should be discussed at the higher level of the whole AI field and together with experts of other disciplines such as jurists. Indeed, the question of liability and responsibility arises in many AI applications, for example when an AI system may perturb stock exchange markets and economy or may endanger or even take the life of passengers of automatic cars. The AI-assisted detection of online grooming is indeed concerned with these questions of unintended usage and liability around AI. However, for a fuller and clearer debate, these questions should be (and are [1-3]) discussed more globally.
>
> [1] Liability for Artificial Intelligence and other emerging digital technologies, European Commission Report from the Expert Group on Liability and New Technologies – New Technologies Formation, 2019
> ISBN 978-92-76-12959-2, doi:10.2838/573689
> https://ec.europa.eu/transparency/regexpert/index.cfm?do=groupDetail.groupMeetingDoc&docid=36608
>
> [2] A. Bertolini, Artificial Intelligence and Civil Liability, Report from the European Parliament's Committee on Legal Affairs, 2020
> https://www.europarl.europa.eu/RegData/etudes/STUD/2020/621926/IPOL_STU(2020)621926_EN.pdf
>
> [3] J.K.C. Kingston, Artificial Intelligence and Legal Liability, International Conference on Innovative Techniques and Applications of Artificial Intelligence, 2016
> https://arxiv.org/ftp/arxiv/papers/1802/1802.07782.pdf

---

> ### Author Response · Authors · 2020-11-16
> **Response to reviewer - part 2**
>
> - *The term "text normalisation" is spread throughout the paper, but is never concretely defined, and is not immediately inferable.*
>
> The definition of text normalisation, provided in the introduction (originally: “Text normalisation methods were proposed to reduce variance and simplify learning”), has been expanded in the revised manuscript to stress that text normalisation is concerned with choice of word and/or spelling.
>
> - *I was not familiar with the "word semantics representation" noun phrase. After reading the paper I am pretty sure that you mean "vector space model", or word embeddings. Why not use these terms that will probably be more familiar to potential readers?*
>
> WSR indeed refers to a word embedding (before LSTM or self-attention layers). With this formulation, we wished to stress the difference with the contextualised word embedding after LSTM or self-attention. The purpose of the embedding (capturing word semantic) is central to the motivation of our modification method, so this name seemed suitable. We have updated the terminology throughout the revised paper, stressing the difference between the embedding for semantic and the contextualised embedding whenever this may be ambiguous.
>
> - *In Figure 1 (left), you wrote Embedding; at the right you wrote WSR. Are these equivalent?*
>
> Thank you for pointing out this possibly confusing term, the terminology has been harmonised to “word embedding” throughout the revised paper.
>
> - *I understand that you are doing classification at the conversation level, but in page 3, in the "Base models" paragraph, you mention that "with the WSR's embedding provided as input to the OG classifier in place of a sentence embedding". In order to classify a conversation, you need a vector representation of it. How is this obtained? In other words, how are you aggregating the contextualized word representations (i.e., the output of the LSTM or the XLNet encoder), into a single vector representation of the conversation? Are you using the last hidden state of the LSTM or a pooling method? Are you using the [CLS] token of XLNet or something else?*
>
> In base model #1, both the last hidden state of the LSTMs, and all previous states, are provided to the attention mechanism in order to create an optimised embedding (which is computed for the full conversation rather than per message, so it is a conversation embedding). This follows the procedure suggested by the authors of the attention mechanism in (Luong et al., 2015). The attention mechanism outputs a conversation embedding of the same size as the LSTM’s hidden state, namely 256. This is now detailed more in the revised paper.
> In base model #2, the last hidden state of the LSTM is used as conversation embedding.
>
> [CLS] is used at the beginning of each conversation for XLNet, and [SEP] is used in between messages of a conversation for both architectures. The use of [CLS] is now explicit in the revised paper.
>
> - *You mentioned that your dataset contains full conversations with an average of 431 messages per conversation. Are all the conversation turns separated by the [SEP] token? What is the average message length? What max input length did you use as a hyperparameter? Did you use the same text input for Model 1 and Model 2?*
>
> Please, kindly refer to Section 4, paragraph “Input to the models” (just before 4.1), where it is explained that all messages are separated by the [SEP] tokens, with no consideration of the speaker’s identity.
>
> Statistics on message lengths etc. were provided in the sup. materials due to lack of space. We added some numbers in the revised paper, and clarified the sup. material tables. Numbers have changed slightly because, in the sup. materials, we initially computed conversation statistics by combining several conversations of two same users. Since different chat sessions / conversations are classified separately, we now provide separate statistics for easier interpretation. There is an average of 215 messages and 1110 words per classified grooming conversation. Average message length is 4.68 words. For non-grooming conversations, there is an average of 13 messages and 104 words per conversation, with an average message length at 7.04 words.
>
> Conversations are truncated to the last 2,000 words. 12 / 8e-5 % of OG / non-OG conversations needed this truncation.
>
> Models 1 and 2, both with and without augmentations, were trained and tested on the exact same text, to allow for fair comparisons.

---

> ### Author Response · Authors · 2020-11-16
> **Response to reviewer - part 3**
>
> - *Saying XLNet is the SoTA for NLP is a false statement (p. 3 second-last paragraph). First of all, NLP encompasses several tasks and there is no single model superior to all the others in every task. Second, XLNet has already been beaten in several tasks. See the Glue benchmark (https://gluebenchmark.com/leaderboard) for a few examples. You mention that XLNet is the SoTA of NLP again in p. 6; sec. 5.1; second paragraph.*
>
> This shortcut was indeed badly phrased, and it has been corrected in the revised manuscript. We meant to say that transformer models, especially XLNet, are the SoTA in many NLP tasks and a popular model, and XLNet is, therefore, a strong baseline to work from and also to compare against.
>
> - *Saying that XLNet iteratively refines word embeddings from a WSR similar to that of LIU et al. (2017) is only tangentially true and is misleading, in my opinion. Recurrent models such as those relying on LSTMs, are profoundly different to those based on transformers such as XLNet.*
>
> This is a misunderstanding, and we have reformulated this paragraph in the revised manuscript to prevent this happening in the future. We did not mean that the refined contextualised embedding of XLNet after the self-attention layers is similar to the word embedding of Liu et al. (2017). We were actually referring to the initial word embedding that captures word semantic similarly to the word embedding of Liu et al. (2017), before any self-attention layer, and independently of the location of the words within sentences. This is the embedding that we work on.
>
> - *I am not sure that there is a clear correspondence between using recurrent models such as LSTMs and better handling of class imbalance. How is a two-layer LSTM going to help handling class imbalance?*
>
> We observed that the original XLNet failed to train satisfactorily on our dataset and always output the same class, with 0.392 F-score. We were able to solve this issue by using an LSTM to further contextualise the word embeddings and extract a conversation embedding. The reason for this behaviour remains to be investigated. It is not the first time that a transformer model is combined with an LSTM, see for example [1] where this approach obtained good results. Using an LSTM to extract a conversation embedding has the extra advantage of allowing the use of our LSTM-based knowledge integration strategies. We now try to better present this decision process in the revised paper.
>
> [1] Ma, Tweets Classification with BERT in the Field of Disaster Management, 2019
> http://web.stanford.edu/class/cs224n/reports/custom/15785631.pdf
>
> - *You mentioned several times that "text normalisation" was a strong point of your contribution, but in the last paragraph of section 4 (p.4) you say that you do not apply "text normalisation". I understand that you might be referring to different kinds of normalisation, but I think this terminology is confusing. I suggest using more precise language to clearly differentiate what you are referring to.*
>
> In the last paragraph of section 4, we describe the text preparation steps that are applied to the text prior to the DNNs. It is correct that we do not apply an external text normalisation to the text before the DNN, since the DNNs include their own text normalisation.
> We have updated the terminology in the revised manuscript to avoid such confusion in the future.
>
> - *What do you mean by "intersection tests"?*
>
> The global support of the RBF allows considering all words for propagating the influence of each pulling operation, rather than operating on a local support which would require identifying those words that are located within each considered neighbourhood. This identification of neighbouring words would typically be done using intersection tests. The description has been amended in the revised paper to avoid the use of this term.
>
> - *I suggest using the word "representations" rather than "coordinates" for referring to the vector representation of a word (p. 4; elastic pulling paragraph).*
>
> The pulling operation is a spatial displacement of the words’ representations within the embedding space. The equations that define this displacement do use coordinates, therefore it is not possible to avoid the use of this word. However, we have amended this paragraph to clearly define the coordinates of a word’s representation in the embedding space.

---

> ### Author Response · Authors · 2020-11-16
> **Response to reviewer - part 4**
>
> - *The correspondence between rows in Table 1 and the strategies discussed in section 4.2, are not immediately apparent.*
>
> Please, kindly refer to the headings of the paragraphs that describe each strategy, both in sections 4.1 and 4.2. These headings have been used in the second column of Table 1.
>
> The stimulating attention and stimulating LSTM input gates strategies have several variants, as detailed in their respective paragraphs. The names of these variants are indicated in bold (and now also underlined) in the paragraphs, and they are used in the third column of Table 1. As indicated in section 5.1, and now also in 4.2, the two variants “supervision” and “excitation” are not mutually exclusive and may be used together, which is indicated by “superv. + excit.” in the third column of Table 1.
>
> - *When not using GloVe embeddings, how did you initialize your word representations?*
>
> Please kindly refer to the first paragraph of Section 5 for all the details on initialisation and training of models. Base model #1, including its word embedding, is trained from random weights on our dataset.
>
> - *You mention in Table 1 that bold are improved results, but improved with respect to what?*
>
> Since the aim of this experiment is to assess the effect of each proposed strategy, the improvements are with respect to not using these augmentation strategies, i.e. with respect to the non-augmented base models (rows 1, 14, and 18 of Table 1).
>
> - *You report precision and recall in Table 2, but not in Table 1. I think it would be valuable to include these in Table 1, rather than the distance reduction and average resulting distance metrics which are only valid for a few strategies.*
>
> Precision and recall for the experiment of Table 1 are provided in the sup. materials. We have edited Table 1 in the revised manuscript to include them.
>
> - *You could check the following papers for more context on combining representations of different word variants*
>
> We thank the reviewer for these suggestions, they are indeed relevant to our study. We have added the most closely relevant paper (due to space constraints) to the related works section of the revised manuscript.
>
> - *Some typos*
>
> Thank you, we have made these corrections.

---

> > ### Comment · AnonReviewer3 · 2020-11-23
> > **Thank you**
> >
> > Thank the authors for their thorough responses. I will reassess my evaluation after re-reading the paper in the coming days.

---

### Official Review · AnonReviewer1 · 2020-10-28
**Combining deep networks and corpus linguistics for online grooming detection**

**Rating:** 6
**Confidence:** 4

**Review:**

This paper presents an approach to natural language processing which integrates corpus linguistics knowledge within deep neural networks (namely, an LSTM-based architecture with attention). The approach is tailored and evaluated on a specific application, namely online grooming detection.

The approach is based on (1) the normalization of word embeddings by exploiting word semantics representations and word variants; (2) the decomposition of conversation analysis to identify subgoals, by exploiting online grooming processes (or phases); (3) the use of attention to modulate the input gate of LSTM cells. Another significant contribution of the paper is a novel corpus, which extends a previous one (PAN2012). The proposed neural architecture presents several variants in otder to incorporate linguistic knowledge within the model, and the paper reports about such an ablation study.

In the experimental evaluation, linguistic knowledge is injected within two base models, namely one based on LSTMs, and the other one on XL-Net. Performance is shown to be improved with respect to the state-of-the-art.

Although the considered task is indeed very important, a weak point of the paper is that it considers a single domain, and the method looks tailored to such domain. The paper asserts that the same methodology could be applied to different scenarios in the domain of chat conversations, but this is not confirmed by the experimental evaluation. For example, what about the process of identification of "variants": is such process hand-made? Would it be possible to have more details on such part?

- Pag. 7, "This may be due to this capturing of language subtleties helping with distinguishing OG conversations..." -> this sentence should probably be rephrased
- Pag. 7, "have same aim" -> "have the same aim"

---

> ### Author Response · Authors · 2020-11-16
> **Response to reviewer**
>
> We thank the reviewer for their insights. We respond next to their raised questions and comments:
>
> *Although the considered task is indeed very important, a weak point of the paper is that it considers a single domain, and the method looks tailored to such domain. The paper asserts that the same methodology could be applied to different scenarios in the domain of chat conversations, but this is not confirmed by the experimental evaluation. For example, what about the process of identification of "variants": is such process hand-made? Would it be possible to have more details on such part?*
>
> As briefly indicated in Section 3, the CL analysis that produced our annotations involves a heavy use of manual analysis by CL experts. Indeed, expert knowledge doesn’t come for free. However, in such multidisciplinary studies, it is often the case that expert CL knowledge already exists.
>
> The word variants that were (manually) identified and used for this study would be largely re-usable in other contexts and applications of analysing chat conversations. Indeed, a significant number of these variants are related to digital language rather than specifically to online grooming. Only a few variants could be more specifically linked to online grooming, where they would aim to minimize the sexual meaning of some terms for example. We reckon that these few samples would not prevent the re-use of the whole set of variants in other applications.
>
> The evaluation of the discriminative aspect of the variants for a given classification can be done easily and automatically, following the procedure described at the beginning of Section 4.1, using empirical occurrences in positive and negative conversations.
>
> In conclusion, the first strategy of integrating knowledge on discriminative word variants into DNNs, is easily reusable in other applications of analysing chat conversations.
>
> The decomposition of a conversation’s aim into subgoals has been the focus of many social science studies. For example, for extreme ideology groups, such as radical right hate speech and radicalisation, a large corpus of works have identified strategies for persuasion / manipulation through conversations [1-5]. This established baseline of knowledge may be used for our second strategy of integrating knowledge into DNNs through decomposing conversations into subgoals.
>
> The identification of frequent 3-word collocates is automated, as described in (Lorenzo-Dus et al., 2016). The association of their occurrences to the identified subgoals is the only task that may require additional manual work.
>
> For these reasons, the prior knowledge integration methods that we present are not only tailored to online grooming detection, but they could also be used for other classification tasks of chat conversation, such as detecting radicalisation for example. The existing CL knowledge of these applications can be exploited to reach this goal in a multidisciplinary context.
>
> This discussion has been provided in the new version of the paper.
>
> [1] Saridakis, I. and Mouka, E. (2020) A corpus study of outgrouping in Greek radical right computer-mediated discourses, Journal of Language Aggression and Conflict, 8: 188 – 231; DOI: https://doi.org/10.1075/jlac.00038.sar
>
> [2] Baker, P. et al (forthcoming, February 2021) The Language of Violent Jihad, Cambridge: Cambridge University Press.
>
> [3] Brindle, A. (2016) The Language of Hate: A Corpus Lingusitic Analysis of White Supremacist Language, London: Routlege.
>
> [4] Nouri, L. & Lorenzo-Dus, N. (2019). Investigating Reclaim Australia and Britain First’s use of social media: Developing a new model of imagined political communities online, Journal for Deradicalization, 18: 1-37.
>
> [5] Lorenzo-Dus, N. & Nouri, L. (2020) The discourse of the US alt-right online – a case study of the Traditionalist Worker Party blog, Critical Discourse Studies, Critical Discourse Studies, DOI: 10.1080/17405904.2019.1708763

---

> > ### Comment · AnonReviewer1 · 2020-11-23
> > **Thanks**
> >
> > I thank the authors for their clarifications.

---

### Official Review · AnonReviewer2 · 2020-10-29
**Integration vs. Extraction**

**Rating:** 5
**Confidence:** 4

**Review:**

Summary

This work proposes the approach of integrating priors into a DNN in the form of Linguistic sub-models that capture characteristics of OG. The authors use the example of the PAN-12 dataset for sexual predators to use information about linguistics behaviour for the grooming phases. The work then goes to highlight the augmentations that are done on baseline DNN models to include these CL characteristics. The authors then go on to show the impact of these augmenations on performance of classification on the PAN-12 dataset.

Questions

- When reading the descriptions of the linguistic sub-models in the DNN, one has the question if we could compare subsets of a DNN that is trained without these explicit sub-models. and may have learned these representations for normalisation etc. vs the integrated CL knowledge.

Could we work to extract interpretable pieces fo the DNN that will then be comparable to the proposed CL augmentations?

The above is important as work on the PAN-12 dataset has tried to reconcile the NLP approach with also understanding the behaviour of sexual predators, so if we can learn how the DNNs are extracting information, we can better create interpretability models that can be more general for NLP + DNNs.

- The work does well to show the gains we get from including these priors. I think we would be better suited if we also understood in the base models, how much of the priors were learnt.

- Please also include a note about some of the ethical considerations when dealing with the PAN-12 dataset and how the data was created.

---

> ### Author Response · Authors · 2020-11-16
> **Response to reviewer**
>
> We thank the reviewer for their insights. We respond in turn to their raised questions and comments:
> - *When reading the descriptions of the linguistic sub-models in the DNN, one has the question if we could compare subsets of a DNN that is trained without these explicit sub-models. and may have learned these representations for normalisation etc. vs the integrated CL knowledge.*
>
> In our ablation studies, we did perform comparisons where no or only part of the CL knowledge was used for enhancing the DNN. These experimental results are shown in Table 1. The improved results when integrating CL knowledge demonstrate that the non-augmented DNNs could not fully discover this knowledge on their own.
>
> For the selective text normalisation, in particular, our experimental results demonstrate that the non-augmented DNNs fail to discover the knowledge on discriminative word variants. Indeed, fine-tuning on our dataset did not allow the DNNs to identify on their own the variants that, at the same time, have the same meanings and are not discriminative of groomer language. These variants were kept separate in the word embeddings, as indicated by the reported distances for non-augmented models in Table 1.
>
> In the revised submission, we have added a discussion on the inability of the two non-augmented DNNs to discover on their own the same level of CL knowledge that we use to augment the models.
>
> - *Could we work to extract interpretable pieces fo the DNN that will then be comparable to the proposed CL augmentations?*
>
> Using machine learning to discover new knowledge is an interesting research field. However, this would be an entirely different study. Here, we are interested in how we can exploit the knowledge that experts already have, in order to augment DNNs and improve their results.
>
> - *The above is important as work on the PAN-12 dataset has tried to reconcile the NLP approach with also understanding the behaviour of sexual predators, so if we can learn how the DNNs are extracting information, we can better create interpretability models that can be more general for NLP + DNNs.*
>
> In our study, we leave the understanding of sexual predators to CL experts, who have published the results of their analysis e.g. in the cited paper (Lorenzo-Dus et al., 2016). However, we demonstrate that augmenting the DNNs using this expert knowledge, in a multidisciplinary research context, does help in improving the interpretability of the model (see Section 5.2 “Visualisation”).
>
> - *The work does well to show the gains we get from including these priors. I think we would be better suited if we also understood in the base models, how much of the priors were learnt.*
>
> As discussed previously, the knowledge on discriminative word variants could not be discovered by the non-augmented DNNs, and this is highlighted in the revised paper.
>
> Testing whether the non-augmented DNNs could identify the existence of sub-goals and the expression of their related contexts is more difficult without introducing this knowledge in the process. As a simple test, in a new experiment, we could visualise the attention energies of the attention module of base model 1, and of the last self attention layer of base model 2, to check whether the model learnt on its own to focus on the contexts of some sub-goals. This new experiment will take a few days (due to current technical issues with our local computation resources), and we will update this response with the results when we get them.
>
> - *Please also include a note about some of the ethical considerations when dealing with the PAN-12 dataset and how the data was created.*
>
> More discussion has been added in Section 3. The data itself being freely available online, its use does not raise any peculiar ethical concern. Its initial collection by the PJ website (we are not involved in this process) was debated and discussed for example in (Chiang & Grant, 2019; Schneevogt et al., 2018), which are cited in our paper. The annotation of the data by CL experts was performed following the method developed in another study (Lorenzo-Dus et al., 2016) and that is cited in our paper. This annotation being mostly manual, due care has been taken to preserve the mental health of the CL experts.

---

> > ### Comment · AnonReviewer2 · 2020-11-23
> > **Thank you for your adjustments**
> >
> > Thank you for the responses and adjustments in the paper.
> >
> > I understand the challenge with rerunning the base models
> >
> > Thank you for the note on the dataset in Section 3.

---

> > > ### Author Response · Authors · 2020-11-23
> > > **Additional test**
> > >
> > > Thank you for your understanding. We now have completed the pending test for base model 1 (the job for base model 2 is still in a waiting queue).
> > >
> > > We have computed the average and std of attention energy at the locations of our labelled instances of OG processes. The values are 0.0009 (0.0002), lower than energy across all conversations at 0.0016 (0.0128). Thus,  the contexts that the model learnt to focus on are not related to our labelled instances of OG processes. This is an indication that the model was not able to discover on its own the sub-goals that the CL analysis of Lorenzo-Dus et al. (2016) identified, and their associated language. This knowledge is therefore an added value for the model, as also demonstrated by the improved results.
> > >
> > > We have added this test and discussion in the updated version of the paper.

---

> > > > ### Author Response · Authors · 2020-11-24
> > > > **Additional test 2**
> > > >
> > > > The pending test for base model 2 is now completed and we have updated the paper with the results, which are similar to those of base model 1: 0.110 (0.072) average (std) energy for instances of OG processes, slightly lower than the energy across all conversations at 0.120 (0.088).

---

### Review · Ethics_Committee · 2020-12-27

**Decision:**

Concerns raised (can publish with adjustment)

**Ethics Review:**

The initial reviewers for this paper flagged two key issues in ethics:
1. On “dealing with the PAN-12 dataset and how the data was created”
2. On “a system that could be used in law enforcement”

On (1):The author’s understanding of data ethics, based on their reply, may be too simplistic.  Just because data is “freely available online” doesn’t automatically mean it’s ethical to use such data.  Combining two or more different datasets could enable new deductions about individuals that have not been previously considered, esp. the impact on privacy expectations.  Further, it could be that a public dataset may have been improperly collected or published in the first place, and the ethics of reusing that data need to be considered.

The authors did not make it clear that any serious ethical consideration was given to how the data was collected or used.  In the paper, the author notes the PAN2012 dataset is a standard corpus for OG detection; but reviewing that reference, there’s no clear indication that data ethics was considered.  However, from the author’s discussions and an online search, there’s also no obvious indication that anything inappropriate has occurred with respect to the datasets used in the study, especially since any deductions gleaned are used only to focus the attention of investigators on certain bits of evidence and not, for instance, in determining guilt or punishment.  The severity of the crimes targeted may tip a balancing test in ethics toward allowing the study, even if the ethical chain of custody has not been definitively established.

Given that the dataset was released in 2012, one would also expect that if these high-profile datasets were ethically problematic, that subsequent discussions and perhaps papers would’ve been published about such concerns.  But from an online search, these concerns were not found.

In summary, (1) does not seem to be a deal-breaking ethical concern.

On (2):The reviewer is correct to be concerned about the use of AI/ML in law enforcement such as discussed in predictive policing. But this study seems to focus on directing the attention of investigators to certain transcripts that may suggest sexual predators are at work.  The human investigators still must evaluate those conversations to see if action is justified, such as to seek a warrant for a wiretap or other search or even for an arrest.

Although the risks are not at the same degree as other policing AI systems, the risks are also not zero.  Merely flagging someone as a “person of interest” could be enough to kick off serious violations of due process and other rights.  For instance, even if an AI alert is not enough evidence for a conviction, it may seem weighty enough to convince a judge to sign a warrant for a wiretap or search.

Thus, accuracy rates (incl. FP and FN) may be relevant in evaluating the risk that this system could be misdirected at innocent parties.  This harm could be unintentional, e.g., by over-trusting the system and putting more weigh on an AI alert that system designers’ had intended.  The system could also be abused and intentionally used to harass or prosecute an innocent party.

Is this problem with the paper serious enough to block its publication on ethical grounds?  Maybe not, given that it can be remedied by the authors, e.g., by providing a clear statement by the system designers that the alerts rendered by the DNN should not be taken as evidence (incl. probable cause and reasonable suspicion) of any kind by itself, including to obtain a search warrant, and that a full ethics review would be needed before deploying such a system.  While it may be true AI and other systems may be abused by law enforcement (just as they could be abused by many other users), that doesn’t absolve the technology designer of the responsibility to mitigate foreseeable issues as much as practical.

Conclusion

As a conference paper, this seems to be an important proof-of-concept toward another tool in the policing toolbox to catch sexual predators—some of the worst criminals that society wants to stop.

Were the system to be actually used in law enforcement, clear policies and limitations need to be established; and contrary to the author’s last statement in the paper, the designer seems central in helping to specify these policies and limitations, as the person closest to a technology system that few or no non-experts can understand.

Another ethical issue not raised by reviewers is whether the study’s authors have attempted to determine that the datasets used are biased against any particular ethnicity, age, etc. given differences and norms in online conversations.  This isn’t necessarily work that needs to be done in the proof-of-concept phase, but certainly it should be examined before fielding the system in the real world.

In summary, while there are ethical concerns with this paper, all things considered, they don’t seem to be enough to prevent publication of this paper

---

### Decision · Program_Chairs · 2021-01-07
**Final Decision**

**Decision:**

Reject

**Comment:**

Most reviewers did not feel that this paper was ready for publication. I thank the authors for answering all the concerns of the reviewers, running new experiments and submitting a revised version, however, this was not not enough to alleviate the reviewers' concerns, notably relating to the handling of the ethical consideration in the writing of the manuscript.